# Correlation between Polymerase Chain Reaction Identification of Iron Acquisition Genes and an Iron-Deficient Incubation Test for *Klebsiella pneumoniae* Isolates from Bovine Mastitis

**DOI:** 10.3390/microorganisms10061138

**Published:** 2022-05-31

**Authors:** Takeshi Tsuka, Soma Kumashiro, Tsubasa Kihara, Toshiko Iida

**Affiliations:** Clinical Veterinary Sciences, Joint Department of Veterinary Medicine, Faculty of Agriculture, Tottori University, 4-101 Koyama-Minami, Tottori 680-8553, Japan; uverworld.t.sid.s@gmail.com (S.K.); kokopendables@gmail.com (T.K.); aratalala@gmail.com (T.I.)

**Keywords:** *Klebsiella pneumoniae*, bovine mastitis, polymerase chain reaction, iron deficient incubation, enterobactin, yersiniabactin

## Abstract

We investigated the correlation between the polymerase chain reaction (PCR) identification of six virulence genes associated with siderophore activation and the iron-uptake system (iron-acquisition genes; *iuc*A, *ent*B, *fep*A, *ybt*S, *psn*, and *kfu*) in mastitis-associated *Klebsiella pneumoniae* (*K. pneumoniae*). The growth of 37 *K. pneumoniae* isolates from the milk of cows with mild mastitis reared on Japanese dairy farms between October 2012 and December 2014 was examined by incubation in an iron-deficient medium. *ent*B-, *fep*A-, or *ybt*S-positive isolates grew significantly better than *ent*B-, *fep*A-, or *ybt*S-negative isolates after incubating in an iron-deficient medium for three days. Interestingly, the growth of isolates with 0 and ≥4 PCR-positive iron-acquisition genes in the iron-deficient medium were significantly different by day 2, while isolates with 2, 3, and ≥4 PCR-positive iron-acquisition genes grew significantly better than those with no PCR-positive iron-acquisition genes by day 3. Based on the correlation between the results of PCR and iron-deficient incubation tests, iron-deficient incubation for three days can be used to estimate the presence or absence of iron-acquisition genes in mastitis-associated *K. pneumoniae*.

## 1. Introduction

Iron is one of the essential nutrients required by bacteria, but it is commonly present at an insufficient level for their growth in their vertebrate hosts [1,2]. Bacteria cannot directly utilize ferric ions (Fe^3+^), a poorly soluble substance that is commonly present in the body cavity [3]. Thus, they can be eliminated by effectively preventing iron acquisition, through using endogenous iron chelators, such as transferrin in serum and lactoferrin in secretory fluids such as milk, as they increase immediately in response to bacterial infection [1]. However, gram-negative bacteria can overcome these host defenses by synthesizing siderophores with an extremely high affinity and specificity for binding to Fe^3+^, which is 10 orders of magnitude higher than those of transferrin and lactoferrin [1,4,5]. The iron-acquisition function of siderophores in mastitis-associated *Klebsiella pneumoniae* (*K. pneumoniae*), which is superior to that of lactoferrin, facilitates long-term survival within the mammary gland [6]. *Klebsiella* species can produce enterobactin, a phenolate siderophore, and aerobactin, a hydroxamate siderophore [1,7]. Yersiniabactin is utilized via the yersiniabactin locus on a high-pathogenicity island in few *K. pneumoniae* strains, whereas the majority of these bacteria can produce enterobactin [5,8]. Coliform bacteria can utilize bovine mastitis-associated siderophores, such as aerobactin, enterobactin, and yersiniabactin, for surviving within the environment and mammary glands and establishing intramammary infection [2,6,9,10,11,12]. Enterobactin assists in intramammary infection [6,10,11], aerobactin contributes to increased mastitis severity in cows [2,9,11], and yersiniabactin seems to prolong the intramammary survival of coliform bacteria, resulting in the chronicity of bovine mastitis [12].

The acquisition virulence genes are usually identified by genetic analysis, which mainly involves polymerase chain reaction (PCR) [2,5,9,13,14]. For PCR tests, a primer comprising a short, single-stranded DNA sequence specific to various virulence genes is used [5,9,13,14]. The *iuc*ABCD gene cluster, which encodes the proteins involved in aerobactin biosynthesis, is commonly detected in hypervirulent *K. pneumoniae* because it is located on the same virulence plasmid that encodes the *rmp*A gene along with the *iut*A gene, which encodes an aerobactin transporter [15,16,17]. Enterobactin-associated gene clusters present in the *K. pneumoniae* chromosome, such as *ent*ABCDEF and *fep*ABCDG, encode its biosynthesis and the production of transport-mediated proteins, respectively [15]. Yersiniabactin-associated genes, presenting in the yersiniabactin locus within the high-pathogenicity island, include *irp*1 and *irp*2 genes, which encode proteins for yersiniabactin biosynthesis; *ybt* and *fyu* genes, which encode the transporters required for yersiniabactin secretion; and the *psn* gene, which encodes the yersiniabactin receptor [5,15,18]. The *kfu* operon encodes an ABC transport system facilitating Fe^3+^ uptake [10,16]. However, previous PCR-based studies on mastitis-associated *K. pneumoniae* isolates have not targeted multiple virulence genes associated with the activation of three siderophore types and the *kfu* iron-uptake system (referred to as iron-acquisition genes throughout this manuscript).

Laboratory tests are frequently used to investigate the various genotypes of coliform bacteria identified using PCR to assess the association between bacterial phenotype and genotype, such as hypermucoviscosity and drug-resistant phenotypes [7,19]. In vitro laboratory tests, such as inoculation, incubation, or culture tests, have previously been used to demonstrate the degree of association with the targeted bacteria [1,3,5,8,14]. Iron-deficient growth media are commonly used in incubation tests involving coliform bacteria and various siderophores to assess their effects on enhancing bacterial survival under iron-restricted conditions [1,3,8]. Additionally, an iron-deficient incubation test has previously been used to evaluate the growth rates of *K. pneumoniae* with or without iron-acquisition genes. Bovine mammary epithelial cell cultures have previously been used to identify the adhesion activity associated with fimbriae genes for the in vitro assessment of coliform isolates from bovine mastitis samples [20]. However, mastitis-associated coliform bacteria have not been incubated in an iron-deficient medium previously. Thus, the design of this study, using incubation in an iron-deficient medium has not been employed previously for *K. pneumoniae* isolates from bovine mastitis. 

This study aimed to present the clinical states of the animals with mastitis induced by the *K. pneumoniae* isolates tested, employ PCR to determine the genes involved in virulence, including six iron-acquisition genes in these *K pneumoniae* isolates, and identify any correlations between the PCR-identified genes and clinical states, as well as between PCR and iron-deficient incubation results. Additionally, previous medical or laboratory reports were examined to determine whether incubating in an iron-deficient medium is a potentially effective in vitro test to estimate the presence of iron-acquisition genes in *K. pneumoniae* isolates and the degree of their association in facilitating bovine mastitis.

## 2. Materials and Methods

### 2.1. Sampling

Thirty-seven abnormal milk samples were obtained between October 2012 and December 2014 from cows with mild mastitis and reared on 18 dairy farms (farms A–R) located in the central area of Tottori prefecture, Japan. Clinical data for the cows included age, days since calving, and parity at the examination of mastitis milk (Appendix A). Additionally, if these animals died by the time of follow-up, the time between their examination and death, and the cause of death were recorded (Appendix A). In this study, the mastitis milk conditions of 37 samples were evaluated by two methods: (1) bacterial counts were determined by performing appropriate 10-fold dilutions of the specimens and expressed as log_10_ colony-forming units (CFU)/mL [21]. (2) The macroscopic appearance of the samples was assigned to one of the four grades according to Gurjar’s method. Score 1: normal, white, homogenous milk; score 2: flake or clot formation in the milk; score 3: a change in the milk appearance to become watery; and score 4: serum- or blood-like changes in the milk [22]. *K. pneumoniae* isolated from the milk specimens using MacConkey–inositol–carbenicillin agar was identified using a VITEK-2 XL Microbiology analyzer (BioMérieux, St. Louis, MO, USA) [23]. The isolates were stored in a skim milk medium at −80 °C until genetic analysis and incubation tests were performed. 

### 2.2. Detection and Identification of Virulence Genes

The frozen specimens were initially thawed, and subsequently cultured in a MacConkey medium. Suspensions were prepared according to the McFarland standard 4 by mixing colonies formed on the medium with sterilized distilled water and incubating at 37 °C for one day. Templates were created by subsequent treatment of the suspensions at 100 °C for 10 min. Primes specific to virulence genes were used for simplex PCR sequencing, as previously described (Appendix A) [5,9,13,14]. After mixing 1 µL template with 9 µL premix comprising 1 μL forward primer, 1 μL reverse primer, 7 μL nuclease free water, and 10 μL green master mix, the solution was initially subjected to 30 amplification cycles comprising denaturation at 94 °C for 2 min, followed by annealing at 50 °C for 45 s, and extension at 72 °C for 60 s. Subsequently, the final extension step was performed at 72 °C for 7 min. The products were separated by electrophoresis using a 1.5% agarose gel supplemented with 0.5 µg/mL ethidium bromide and TAE buffer at 100 V. The presence of target genes was confirmed when amplicons with expected sizes were detected.

### 2.3. Incubation Tests 

The experiments were performed in a double-blind format in which one individual (S.K.) performed the isolation tests, without knowing the PCR results that had been sequenced by other individuals (T.K., and T.I.). The iron-deficient medium used in this study was designed according to what has been reported in previous studies [24,25]. The basic medium solution comprised 35 mM glucose, 25 mM NH_4_Cl, 1.5 mM KCl, 45.2 mM NaCl, and 0.4 mM MgSO_4_·7H_2_O, supplemented with 0.5 µM CaCl_2_·6H_2_O, 0.5 µM H_3_BO_3_, 0.05 µM CoCl_2_·6H_2_O, 0.05 µM CuSO_4_·7H_2_O, 0.05 µM ZnSO_4_·7H_2_O, 0.1 µM MnSO_4_, and 0.005 µM (NH_4_)6Mo_7_O_24_·4H_2_O. These components were diluted with 66.7 mM sodium phosphate buffer (pH 7.4) and then adjusted using a mixture of Na_2_HPO_4_ and NaH_2_PO_4_. The iron-deficient medium was prepared by batch incubating the solution for one day with Chelex 100 ion-exchange resin (Bio-Rad Laboratories, Inc., Hercules, CA, USA) to remove any iron. Iron-half-sufficient and iron-sufficient media were produced by supplementing the iron-deficient medium with 10 and 20 µM FeSO_4_·7H_2_O, respectively.

*K. pneumoniae* solutions were prepared according to the McFarland standard 3 by mixing the isolates with sterilized distilled water, naturally thawing the frozen specimens, and incubating in a MacConkey medium at 37 °C for one day. *K. pneumoniae* solutions were diluted by mixing 10 µL bacterial solution with 990 µL pre-prepared substrate solutions (one of the three). *K. pneumoniae* isolates were incubated in an iron-sufficient, iron-half-sufficient, or iron-deficient medium at 37 °C for 1, 2, and 3 days. The bacterial counts were measured using the serial dilution technique with 100 µL aliquots collected from each medium daily. 

### 2.4. Statistical Evaluation

#### 2.4.1. Clinical Data and Mastitis Milk Conditions

In this study, *K. pneumoniae* isolates were grouped as KP0, KP1, KP2, KP3, and KP4 based on the number of iron-acquisition genes (0, 1, 2, 3, and ≥4, respectively), as identified by PCR tests. The Kruskal–Wallis test was used for statistical comparison among these KP groups and total values in terms of the clinical data and milk conditions in the 37 mastitis cases.

#### 2.4.2. Comparison between Inoculation Tests and PCR Identification

In this study, the *K. pneumoniae* counts on days 0, 1, 2, and 3 in iron-sufficient and iron-deficient media were designated as KC0_IS_, KC1_IS_, KC2_IS_ and KC3_I__S_, and KC0_ID_, and KC1_ID_, KC2_ID_ and KC3_ID_, respectively. One-way analysis of variance (one-way ANOVA) and Kruskal–Wallis tests were used to compare the positive and negative PCR results for iron-acquisition genes (*ent*B, *fep*A, *ybt*S, *psn*, and *kfu*) with respect to the association with the bacterial counts on incubation day (0–3 days) in the three media. Additionally, a Kruskal–Wallis test was used to compare the bacterial counts in iron-sufficient, iron-half-sufficient, and iron-deficient media on days 0, 1, 2, and 3. 

#### 2.4.3. Association between >8 log_10_ CFU/mL KC3_ID_ and PCR-Positive Combination of Two Iron-Acquisition Genes

In this study, >8 log_10_ CFU/mL KC3_ID_ was used as the basis for evaluating *K. pneumoniae* survival under iron-deficient conditions as it was identified as the common count on incubation day 3, in an iron-sufficient medium. In this analysis, PCR-positive combinations of two of five iron-acquisition genes (*ent*B, *fep*A, *ybt*S, *psn*, and *kfu*) were evaluated. The support, confidence, and lift values were analyzed using association analysis, based on a previously reported statistical analysis of the diagnostic role of milk [26]. In this previous statistical method, the degrees of association were estimated to be strong when the lift values were >1, as a minimum positive dependence effect [26]. Additionally, the chi-square test was used to compare the isolates with each PCR-positive combination and total 37 isolates. 

#### 2.4.4. Comparison between Incubation Tests and Numbers of Iron-Acquisition Genes According to the PCR Results

This statistical analysis was conducted for each iron-sufficient, iron-half-sufficient, and iron-deficient incubation test. One-way ANOVA or the Kruskal–Wallis test was used to compare the bacterial counts on days 0, 1, 2, and 3 for each KP group (KP0, KP1, KP2, KP3, and KP4). The Kruskal–Wallis test was used to compare the daily bacterial counts for the various KP groups.

The Scheffe and Mann–Whitney U tests were used for post hoc analysis after the one-way ANOVA and Kruskal–Wallis test throughout this study, respectively. A *p*-value of <0.05 was considered statistically significant.

## 3. Results

### 3.1. PCR Identification

PCR did not detect *mag*A or *rmp*A genes associated with the hypermucoviscosity phenotype, or the K1 and K2 capsular polysaccharide genes in any of the *K. pneumoniae* isolates. In terms of the PCR identification of the six iron-acquisition genes, the *iuc*A gene was not detected in any of the 37 *K. pneumoniae* isolates (Appendix A). *ent*B and *fep*A genes were most commonly detected using PCR (both, 67.6%; Appendix A). The average number of PCR-positive iron-acquisition genes in the isolates was 2.6 (Appendix A). 

### 3.2. Clinical Data and Mastitis Milk Conditions

The numbers (proportions) of KP0, KP1, KP2, KP3, and KP4 isolates were 6 (16.2%), 6 (16.2%), 14 (37.8%), 6 (16.2%), and 5 (13.5%), respectively (Table 1). The clinical data and milk conditions were not significantly different among the KP groups and the 37 isolates. 

### 3.3. Comparison between Inoculation Tests and PCR Identification

KC1_IS_ was significantly (*p* < 0.05) higher than KC0_IS_ and similar to KC2_IS_ and KC3_IS_, which is similar to the growth of PCR-positive and PCR-negative *K. pneumoniae* isolates with each iron-acquisition gene in the iron-sufficient medium (Table 2). Moreover, the KC3_IS_ of *ent*B-positive, *fep*A-positive, *psn*-positive, and *kfu*-negative *K. pneumoniae* isolates was significantly (*p* < 0.05) higher than their KC1_IS_. The growth pattern of *ent*B-positive *K. pneumoniae* isolates in the iron-deficient medium was similar to that in the iron-sufficient medium. However, the growth pattern of *ent*B-negative *K. pneumoniae* isolates in the iron-deficient medium showed that the KC1_ID_ (7.67 ± 1.08 log_10_ CFU/mL) was significantly (*p* < 0.05) higher than the KC0_ID_ (5.59 ± 0.42 log_10_ CFU/mL), KC2_ID_ (6.42 ± 1.13 log_10_ CFU/mL), and KC3_ID_ (5.89 ± 1.60 log_10_ CFU/mL). Thus, the KC3_ID_ was not significantly different from the KC0_ID_. Compared with the iron-deficient growth patterns between *ent*B-negative and *ent*B-positive isolates, significant (*p* < 0.05) differences were found in the KC2_ID_ (6.42 ± 1.13 and 7.72 ± 0.65 log_10_ CFU/mL, respectively), and the KC3_ID_ (5.89 ± 1.60 and 7.63 ± 0.94 log_10_ CFU/mL, respectively).

The growth pattern of *ybt*S-positive and *ent*B-positive isolates in the iron-deficient medium were similar; thus, the KC3_ID_ of *ybt*S-positive and *ybt*S-negative isolates were significantly (*p* < 0.05) different (8.43 ± 0.18 and 6.90 ± 1.42 log_10_ CFU/mL, respectively). The iron-deficient growth patterns of *fep*A-negative, *ybt*S-negative, *psn*-negative, and *kfu*-negative *K. pneumoniae* isolates indicate a gradual decrease from KC2_ID_ to KC3_ID_ subsequent to the increase between KC0_ID_ and KC1_ID_. In the *fep*A-negative isolates, the KC1_ID_ (7.56 ± 1.06 log_10_ CFU/mL) was significantly (*p* < 0.05) higher than the KC0_ID_ (5.64 ± 0.39 log_10_ CFU/mL), and KC3_ID_ (6.07 ± 1.60 log_10_ CFU/mL). Additionally, the KC3_ID_ of *fep*A-positive and *fep*A-negative isolates were significantly (*p* < 0.05) different (7.54 ± 1.08 and 6.07 ± 1.60 log_10_ CFU/mL, respectively). In *ybt*S-negative, *psn*-negative, and *kfu*-negative isolates, the KC0_ID_ were significantly (*p* < 0.05) lower than the KC1_ID_, KC2_ID_, and KC3_ID_. The iron-half-sufficient growth patterns tended to be similar to the iron-sufficient growth patterns (Appendix A).

### 3.4. Association between the KC3_ID_ of >8 log_10_ CFU/mL and PCR-Positive Combination of Two Iron-Acquisition Genes

The most common combination of two PCR-positive iron-acquisition genes related to >8 log_10_ CFU/mL KC3_ID_ comprised *ent*B and *fep*A genes (support value: 0.30), but the lift value (1.02) was close to 1 (implying a minimum positive dependence effect; Table 3). The combination of PCR-positive *ybt*S and *kfu* genes contributed to >8 log_10_ CFU/mL KC3_ID_, with the highest lift value (12.33) but the lowest support value (0.08). In this analysis, many *K. pneumoniae* isolates with the PCR-positive *ybt*S gene, as one of two iron-acquisition genes, could be grown at >8 log_10_ CFU/mL in the iron-deficient medium within three days (Table 3).

### 3.5. Comparison between Incubation Tests and PCR-Positive Numbers of Iron-Acquisition Genes

The growth patterns of KP1, KP2, KP3, and KP4 in iron-sufficient medium for 0–3 days were similar, and the KC0_IS_ of all KP groups were significantly (*p* < 0.05) lower than their KC1_IS_, KC2_IS_, and KC3_IS_, except there was no significant difference between KC0_IS_ and KC1_IS_ in KP0 (Figure 1). 

The KC1_ID_, KC2_ID_, and KC3_ID_ of KP2, KP3, and KP4 isolates were significantly (*p* < 0.05) increased compared to their KC0_ID_ (Figure 2). The values tended to be higher in an order dependent on the number of PCR-positive iron-acquisition genes in KC1_ID_, KC2_ID_, and KC3_ID_. The KC1_ID_ of KP0 isolates were significantly (*p* < 0.05) higher than their KC0_ID_, followed by gradual decreases in KC2_ID_ and KC3_ID_. However, the KC2_ID_ and KC3_ID_ of KP0 isolates were not significantly different from their KC0_ID_. Interestingly, the KC2_ID_ of KP0 and KP4 isolates were significantly (*p* < 0.05) different. Furthermore, the KC3_ID_ of KP0 isolates was significantly (*p* < 0.05) lower than that of KP2, KP3, and KP4 isolates. The growth pattern of KP1 isolates was similar to that of KP0 isolates, where the significant (*p* < 0.05) increase between KC0_ID_ and KC1_ID_ was followed by a gradual decrease from KC1_ID_ to KC3_ID_. Interestingly, the KC3_ID_ of KP1 isolates was significantly (*p* < 0.05) lower than that of KP4 isolates. The growth patterns associated with the variation in number of PCR-positive iron-acquisition genes in iron-half-sufficient medium were similar to those in the iron-sufficient medium (Appendix A). 

## 4. Discussion

This study identified the variations in the number of PCR-positive iron-acquisition genes in *K. pneumoniae* isolates between farms. Several genotypes of mastitis-associated *K. pneumoniae* have frequently been detected [27,28], because multiple genotypes of this pathogen are commonly present in sawdust bedding and the feces of cows reared in each farm [6]. The extended intra-farm distribution of *K. pneumoniae* might have contributed to the variety of PCR-positive iron-acquisition genes in this study. The average age and parity of the 37 cows used in this study were 5.5 years and 3.5, respectively, with the average mastitis duration being 162 days after calving. These data are similar to previously reported findings, where mild mastitis occurred during 146 lactation days in 4.1 years-old cows with 2.7 average parity [21]. Additionally, the prevalence of *Klebsiella*-associated mastitis has previously accounted for approximately 30% at >100 days after calving, although approximately 50% at <30 days after calving [29]. The bacterial counts in the milk of cows with mild mastitis in this study were 5.9 log_10_ CFU/mL, which is within the previously identified level (>5.0 log_10_ CFU/mL) in >25% of mild mastitis milks, although previous reports have also shown that 6.0 log_10_ CFU/mL develops acute clinical signs of coliform mastitis [21,29]. However, the clinical data of *K. pneumoniae*-infected cows with mastitis were not related to *K. pneumoniae* identification using PCR. 

PCR analysis of 37 *K. pneumoniae* isolates did not detect the *rmp*A and *mag*A genes related to the hypermucoviscosity phenotype and the *iuc*A gene, which is related to aerobactin biosynthesis and found on the same virulence plasmid, in any isolate [9,15,16,17]. Hypervirulent *K. pneumoniae* strains with the virulence plasmid encoding these genes exhibit predominant aerobactin production-associated siderophore activity, in contrast to the reduced enterobactin and yersiniabactin-associated activity [15,16,17,19]. Previous PCR analyses of *K. pneumoniae* isolated from mastitis milk found that the prevalence of *iuc*A-positive isolates ranged between 66.7% and 100% [2,9]. Additionally, the hypermucoviscosity phenotype associated with *rmp*A and *mag*A expression was found in 16% of *K. pneumoniae* isolates from bovine mastitis samples [16]. In contrast, the PCR-negative *K. pneumoniae* isolates for the *iuc*A, *rmp*A, and *mag*A genes in this study seemed to belong to the classical *K. pneumoniae* strains, which distinguished them from hypervirulent *K. pneumoniae* strains [7]. Therefore, classical *K. pneumoniae* strains may have been one of the possible causes of the present bovine cases, as they involved mild mastitis. A previous study has predominantly detected aerobactin in *K. pneumoniae* isolates from animals with moderate to severe clinical mastitis using PCR [11]. 

Interestingly, the growth pattern of *K. pneumoniae* isolates in the iron-deficient medium was identical to the common patterns in iron-sufficient incubation, even though these isolates were PCR-positive for only one of the six iron-acquisition genes analyzed in this study. In particular, the growth of *ent*B-positive isolates in the iron-deficient medium was significantly enhanced between days 2 and 3. Enterobactin, which is biosynthesized by an *ent*B gene-encoded protein, is one of the most common siderophores secreted by *K. pneumoniae* when it infects the mammary glands, facilitating increased intramammary colonization [6,10,11]. The KC3_ID_ of isolates that were PCR-positive and PCR-negative for *ybt*S and *fep*A were also significantly different. Yersiniabactin, which is biosynthesized by a *ybt*S gene-encoded protein, may promote chronicity of bovine mastitis via prolonged survival within infected mammary glands because of its function in biofilm formation as well as the iron-acquisition system in iron-deficient environments [12]. Mastitis-associated coliform bacteria predominantly possess the *fep*A gene, the activity of which is necessary for interacting with the enterobactin-mediated iron retrieval system on the cellular surface [6]. The *kfu*- and *psn*-positive *K. pneumoniae* strains seemed to minimize the iron-deficient-associated decrease between KC2_ID_ and KC3_ID_. The association of the *psn* gene with bovine mastitis is not well known, because there are no previous reports about this association, while the *kfu* iron-acquisition system is assumed to play a common role in promoting intramammary infection by mastitis-associated *K. pneumoniae*; the prevalence of *kfu*-positive *K. pneumoniae* accounts for 25% of caprine mastitis and 77.8% of bovine and buffalo mastitis [2,13,16]. Additionally, the *kfu* iron-acquisition system may enhance bovine mastitis severity based on the association of *kfu* genes with subclinical and clinical bovine mastitis prevalence (20% vs. 39%, respectively) [10].

In addition to evaluating single iron-acquisition genes, assessing the associations between the number of PCR-positive genes and combinations of multiple iron-acquisition genes from the results of iron-deficiency incubation tests is useful because interactions between various iron-acquisition systems may enhance the bacterial potential to adapt to iron-deficient environments [2,7,8,14,17]. Accordingly, the combination of *ent*B and *fep*A genes was most commonly associated with >8 log_10_ CFU/mL KC3_ID_. In mastitis-associated *K. pneumoniae* isolated in our field study, the enterobactin-mediated iron-acquisition system enhanced by the yersiniabactin-mediated system may contribute to prolonged survival within the iron-deficient conditions of infected mammary glands [2,14,15]. Additionally, *ent*B gene expression followed by *ybt*S gene expression promotes siderophore activation in 46.3–83.7% *K. pneumoniae* isolates with these genes [17]. However, the highest association of this combination with this criterion was simply caused by the highest PCR-positive proportion of this combination (54.1%) compared to those of the other combinations (8.1–29.7%). Based on the lift values, the combination of the *ybt*S gene with the other genes had high contributions for this criterion despite their low PCR-positivity (8.1–13.5%). The previous PCR analysis for classical *K. pneumoniae* strains related to human infectious diseases identified that the PCR-positivity of the *ybt*S gene was lower than that of the *ent*B gene [30]. Co-activating the genes encoding siderophore transport or receptor systems, together with their biosynthesis genes, such as *ent*B and *ybt*S, may influence the enhanced pathogenicity of some strains [8]. Co-expression of *ybt* and *psn* genes may be induced in *K. pneumoniae* with a high-pathogenicity island locus encoding these genes under iron-deficient conditions [5,15,18]. However, no previous reports have focused on the role of enhanced yersiniabactin-mediated pathogenicity resulting from co-activation of the *ybt*S and *psn* genes associated with bovine mastitis. The increased association of the combination of *ybt*S and *kfu* genes with bovine mastitis may be supported by previous inoculation tests identifying the significant effects of decreasing the lethal doses in mice infected with *K. pneumoniae* strains with these two genes, possibly facilitating their growth in host animals [8].

All five KP4 isolates in this study could utilize enterobactin and yersiniabactin, and four of the five isolates might have the *kfu* iron-acquisition system. *K. pneumoniae* strains with multiple iron-acquisition systems account for >90% of pathogenic *K. pneumoniae* strains in humans [17]. Virulence genes encoding enterobactin, aerobactin, and yersiniabactin have also frequently been detected in PCR analyses of *K. pneumoniae* isolates from environmental samples collected from dairy farms, as well as mastitis milk samples [2,11,20]. Coliform bacteria, including *K. pneumoniae*, can profitably utilize several iron chelators when invading and subsequently surviving within the mammary glands due to the selective and effective diversion of these acquired substances for their survival in the dairy environment [2]. Intramammary infections caused by these bacteria can facilitate frequent occurrences of mild or persistent mastitis with acute clinical signs [2,31,32]. Interestingly, no predominant virulence gene combination impacts the severity of *Escherichia coli-*associated mastitis [33]. The relationship of our data with the severity of *K. pneumoniae*-associated mastitis should be further assessed, because isolates from mild mastitis cases were targeted in this study.

Our study highlighted the utilization of iron-deficient incubation to assess correlations with PCR identification. Time-dependent changes in the incubation tests appeared to differ from those reported in a previous study, showing a sharp increase in *K. pneumoniae* counts (6–8 log_10_ CFU/mL) within 6 h, followed by a plateau at the level of 8 log_10_ CFU/mL between 6 and 24 h [4]. A previous study has reported that *K. pneumoniae* with various iron-acquisition genes grow rapidly up to 8 h, followed by a gradual increase between 8 and 24 h, regardless of the intra-medium iron concentrations (0, 10, 30, and 50 µM), corresponding to the levels between the iron-deficient and iron-sufficient media used in this study [3]. Contrary to previous growth changes within 24 h [3,4], our preliminary experiments have reported the continuous, rapidly increasing growth of *K. pneumoniae* incubated in an iron-sufficient medium for 24 h (data not shown). During this incubation period, *K. pneumoniae* may utilize Fe^3+^ ions stored within the cells, regardless of the genes identified by PCR. Based on the preliminary tests, incubation tests were designed to be performed over three days. The results on the third day enabled us to identify the differences among variations in the PCR identifications of iron-acquisition genes. On the third day of incubation, the counts of *K. pneumoniae* with no PCR-detected iron-acquisition genes might have naturally decreased, because the death rates exceeded their growth rates due to Fe^3+^ ion depletion. *K. pneumoniae* may be more efficient for acquiring poor Fe^3+^ ion concentration, dependent upon the increased number of PCR-positive iron-acquisition genes. The iron concentration in the iron-deficient medium described previously and used in this study is less than 0.5 µM [3,25]. Our study using iron-deficient incubation tests can be developed by including other laboratory tests, such as bovine mammary epithelial cell culture to identify bacterial adhesion function, for the further investigation of the virulence genes facilitating *K. pneumoniae*-associated bovine mastitis [20].

## 5. Conclusions

In this study, PCR analysis of six iron-acquisition genes (*iuc*A, *ent*B, *fep*A, *ybt*S, *psn*, and *kfu*) using 37 *K. pneumoniae* isolates from bovine mastitis milk confirmed the higher proportions of PCR detection for *ent*B and *fep*A genes, contrary to the lack of PCR detection for the *iuc*A gene. The growth pattern of *K. pneumoniae* isolates that were PCR-positive for each iron-acquisition gene in the iron-deficient medium were identical to those in the iron-sufficient medium. The count of mastitis-associated *K. pneumoniae* isolates that were PCR-positive for *ybt*S and the other iron-acquisition genes (*ent*B, *fepA*, *psn*, and *kfu*) incubated in iron-deficient and iron-sufficient media were mostly >8 log_10_ CFU/mL, as a level of three-day incubation. Moreover, *K. pneumoniae* isolates with four and five PCR-positive iron-acquisition genes could grow in the iron-deficient medium for three days compared to those with 0 and 1 PCR-positive iron-acquisition genes. This iron-deficient incubation test is so simple that it is routinely applicable in laboratories in bovine practice, which are not commonly equipped with advanced examination devices (such as a PCR analyzer). Evaluating iron-deficient incubation growth using 8 log_10_ CFU/mL KC3_ID_ may contribute to estimating the degree of iron-acquisition function in mastitis-associated *K. pneumoniae* the need to perform PCR analysis.

## Figures and Tables

**Figure 1 microorganisms-10-01138-f001:**
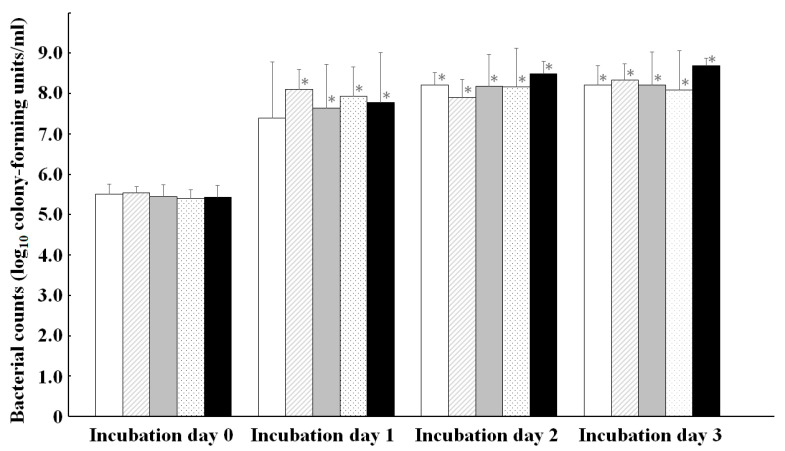
Correlations among *Klebsiella pneumoniae* isolates grouped by KP0 (white), KP1 (diagonal), KP2 (gray), KP3 (dotted), and KP4 (black) for the growths in iron-sufficient medium. * Asterisks denote significant (*p* < 0.05) differences for each value of KP groups on incubation day 0.

**Figure 2 microorganisms-10-01138-f002:**
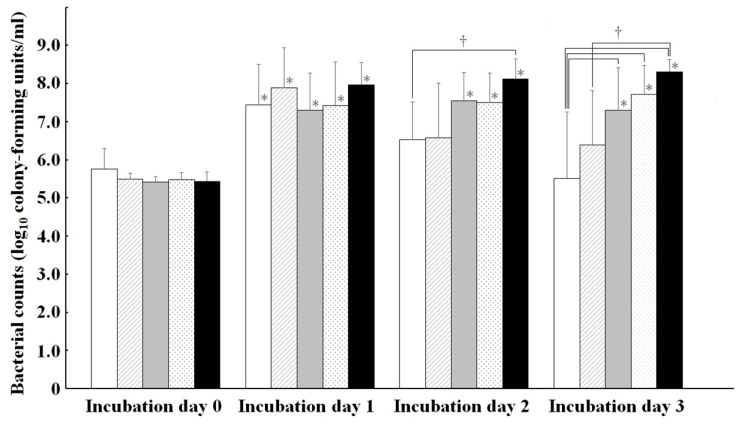
Correlations among *Klebsiella pneumoniae* isolates grouped by KP0 (white), KP1 (diagonal), KP2 (gray), KP3 (dotted), and KP4 (black) for the growths in iron-deficient medium. * Asterisks denote significant (*p* < 0.05) differences for each value of KP groups on incubation day 0. † Dagger symbols denote significant (*p* < 0.05) differences between KP groups, connected by a line.

**Table 1 microorganisms-10-01138-t001:** Comparison of clinical data [mean (standard deviation)] in mastitis cases, and mastitis milk conditions associated with polymerase-chain-reaction (PCR)-positive numbers of iron-acquisition genes in *Klebsiella pneumoniae* isolates.

KP Group ^1^	Number [Proportion]	Age (days) at Exam	Days since Calving at Exam	Parity at Exam	Interval (Days) between Exam and Death	Mastitis Milk Conditions	
						Bacterial Counts (log_10_ CFU/mL)	Score [20]
KP0	6 [16.2%]	2009.0 (757.7)	220.5 (244.7)	2.8 (1.3)	872.5 (754.1)	5.9 (2.2)	2.5 (1.2)
KP1	6 [16.2%]	1776.4 (823.6)	137.8 (103.5)	3.2 (2.0)	596.2 (529.6)	7.0 (1.5)	2.0 (0.6)
KP2	14 [37.8%]	2056.2 (757.4)	203.3 (145.1)	3.6 (1.7)	408.4 (346.3)	5.4 (2.1)	2.3 (1.0)
KP3	6 [16.2%]	2251.8 (536.9)	74.0 (102.5)	4.6 (1.3)	632.0 (636.0)	6.5 (2.3)	2.6 (1.1)
KP4	5 [13.5%]	1809.6 (718.0)	93.5 (112.3)	3.0 (1.8)	893.5 (464.9)	5.0 (1.3)	3.3 (1.0)
Total	37 [100.0%]	2000.9 (708.1)	162.3 (156.1)	3.5 (1.7)	620.0 (528.2)	5.9 (2.0)	2.4 (1.0)

^1^ PCR-positive numbers (0, 1, 2, 3, and >4) of iron-acquisition genes in *Klebsiella pneumoniae* isolates (KP0, KP1, KP2, KP3, and KP4, respectively).

**Table 2 microorganisms-10-01138-t002:** Association between bacterial counts [means (standard deviations); log_10_ colony-forming units/mL] and polymerase-chain-reaction (PCR)-positive iron-acquisition genes in *Klebsiella pneumoniae* isolates incubated in iron-sufficient and iron-deficient medium.

Gene	PCR ^1^	Iron-Sufficient Incubation				Iron-Deficient Incubation			
		Day 0	Day 1	Day 2	Day 3	Day 0	Day 1	Day 2	Day 3
*ent*B	+	5.44 (0.26) ^a^	7.73 (1.00) ^b,c^	8.30 (0.65) ^b,d^	8.37 (0.70) ^b,d^	5.45 (0.17) ^a^	7.42 (0.93) ^b^	7.72 (0.65) ^b,e^	7.63 (0.94) ^b,e^
*ent*B	−	5.50 (0.20) ^a^	7.74 (1.10) ^b^	7.93 (0.63) ^b^	8.08 (0.61) ^b^	5.59 (0.42) ^a^	7.67 (1.08) ^b,c^	6.42 (1.13) ^b,d,f^	5.89 (1.60) ^d,f,g^
*fep*A	+	5.44 (0.26) ^a^	7.75 (0.97) ^b,c^	8.23 (0.66) ^b^	8.34 (0.71) ^b,d^	5.42 (0.17) ^a^	7.47 (0.96) ^b^	7.39 (1.02) ^b^	7.54 (1.08) ^b,e^
*fep*A	−	5.50 (0.20) ^a^	7.68 (1.17) ^b^	8.07 (0.65) ^b^	8.14 (0.62) ^b^	5.64 (0.39) ^a^	7.56 (1.06) ^b,c^	7.04 (1.09) ^b^	6.07 (1.60) ^b,d,f^
*ybt*S	+	5.41 (0.33) ^a^	7.58 (1.32) ^b^	8.53 (0.34) ^b^	8.70 (0.22) ^b^	5.38 (0.25) ^a^	7.81 (0.63) ^b^	8.20 (0.57) ^b^	8.43 (0.18) ^b,e,g^
*ybt*S	−	5.47 (0.24) ^a^	7.75 (1.00) ^b^	8.14 (0.68) ^b^	8.22 (0.70) ^b^	5.51 (0.28) ^a^	7.47 (1.00) ^b^	7.16 (1.03) ^b^	6.90 (1.42) ^b,f^
*psn*	+	5.47 (0.24) ^a^	7.89 (1.08) ^b,c^	8.13 (0.82) ^b^	8.28 (0.74) ^b,d^	5.46 (0.20) ^a^	7.66 (0.92) ^b^	7.84 (0.79) ^b^	7.35 (1.38) ^b^
*psn*	−	5.46 (0.25) ^a^	7.68 (1.01) ^b^	8.19 (0.62) ^b^	8.27 (0.68) ^b^	5.51 (0.30) ^a^	7.46 (1.00) ^b^	7.14 (1.05) ^b^	6.99 (1.45) ^b^
*kfu*	+	5.37 (0.20) ^a^	7.99 (0.87) ^b^	8.15 (0.84) ^b^	8.18 (0.83) ^b^	5.47 (0.18) ^a^	7.83 (1.04) ^b^	7.64 (0.80) ^b^	7.78 (0.89) ^b^
*kfu*	−	5.51 (0.26) ^a^	7.58 (1.08) ^b,c^	8.20 (0.55) ^b^	8.32 (0.60) ^b,d^	5.51 (0.32) ^a^	7.33 (0.92) ^b^	7.06 (1.11) ^b^	6.68 (1.53) ^b^
Total		5.46 (0.24)^a^	7.73 (1.02) ^b,c^	8.18 (0.66) ^b^	8.27 (0.68) ^b,d^	5.49 (0.28) ^a^	7.50 (0.98) ^b^	7.28 (1.04) ^b^	7.07 (1.43) ^b,h^

^1^ + and −, Each iron-acquisition gene was detected or not detected by the PCR test, respectively. ^a,b,c,d^ Within row (among 0–3 incubation days) for each incubation group, numbers with different superscripts are significantly different (*p* < 0.05). ^e,f^ Within column (between PCR^+^ and PCR^−^) for each iron-acquisition gene group, numbers with different superscripts are significantly different (*p* < 0.05). ^g,h^ Within column between each iron-acquisition gene group and total values, numbers with different superscripts are significantly different (*p* < 0.05).

**Table 3 microorganisms-10-01138-t003:** Proportion (number) of polymerase-chain-reaction (PCR)-positive combinations of two iron-acquisition genes, and the association analysis ^1^ between the combination and the measurements of *Klebsiella pneumoniae* counts of >8 log_10_ colony-forming units/mL on three incubation days in iron-deficient medium (basic KC3_ID_).

Combination of Two Genes Listed within Column and Row	*fep*A	*ybt*S	*psn*	*kfu*
*ent*B	54.1% (*n* = 20) S: 0.30, C: 0.55, L: 1.02	10.8% (*n* = 4) S: 0.11, C: 1.00, L: 9.25 *	18.9% (*n* = 7) S: 0.11, C: 0.57, L: 3.02	29.7% (*n* = 11) S: 0.19, C: 0.64, L: 2.14
*fep*A	-	10.8% (*n* = 4) S: 0.11, C: 1.00, L: 9.25 *	13.5% (*n* = 5) S: 0.11, C: 0.80, L: 5.92 *	29.7% (*n* = 11) S: 0.16, C: 0.55, L: 1.83
*ybt*S	-	-	10.8% (*n* = 4) S: 0.11, C: 1.00, L: 9.25 *	8.1% (*n* = 3) S: 0.08, C: 1.00, L: 12.33 *
*psn*	-	-	-	13.5% (*n* = 5) S: 0.08, C: 0.60, L: 4.44

^1^ Support, confidence, and lift values are abbreviated as “S”, “C”, and “L”, respectively, in this table. * Associations between PCR-positive combinations of two genes for the achievement of basic KC3_ID_ are significant (*p* < 0.05) using chi-square test.

## Data Availability

The data used to support the findings of this study are available from the corresponding author upon request.

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
