# Peer review of "Correlation between Polymerase Chain Reaction Identification of Iron Acquisition Genes and an Iron-Deficient Incubation Test for Klebsiella pneumoniae Isolates from Bovine Mastitis"

_microorganisms, 2022, doi:10.3390/microorganisms10061138_

Round 1
Reviewer 1 Report
Tsuka et al. reported on the correlation between six virulent genes identified by PCR and the iron-incubation tests for Klebsiella pneumoniae isolated from bovine mastitis in Tottori region, Japan. Unfortunately, I could not find the reason to publish this manuscript to the novel international journal since this manuscript must be improved massively.
- The number of samples. If this study was used the isolates from humans, it might be worthy. However, the authors used only 37 K. pneumoniae samples isolated from bovine in Tottori region, Japan.
- It requires extensive editing of English language and style required.
Author Response
Thank you for your great advice. During our study period, we could examine approximately 400 milk specimens from mastitis cows rearing in Japanese dairy farms per one year. The isolation prevalence of Klebsiella pneumoniae was approximately 4-5% of these mastitis milks. Therefore, we could get only 37 isolates during our study periods, although Reviewer 1 pointed out about the small number of the specimens for our experiments. In terms of sampling numbers of Klebsiella pneumoniae in previous bovine reports, the levels of hundreds of this bacteria are very difficult to be tested, and the levels of tens could have been frequently used in many bovine previous reports. We would appreciate it if you could consider this background.
The revised version is corrected by English-proofreading company (Editage). Yellow-boxes highlight the corrected parts according to Reviewers’ advise, and by English-proofreading company.
Reviewer 2 Report
Bovine mastitis is the most common disease of dairy cattle. One of the etiological factors is Klebsiella pneumoniae and these strains may have varying profiles of virulence. K.pneumoniae has an absolute requirement for iron for infection success. The growth of bacteria in the iron-rich and iron-restricted mediums was examined. The authors have found a correlation between the composition of iron-acquisition genes (by PCR detection) and the growth of K. pneumoniae in the iron-deficient medium.
The work is careful and well-documented, but the methodology used to testing of virulence is slightly outdated. The incubation test using an iron-deficient medium to estimate the carrying of iron-acquisition genes in bacteria and bacterial counts was known before the q/real-time PCR and CAS liquid assay era, these tests should be also used as a comparative method.
Minor remarks:
1. In the Introduction section, the bovine mastitis problem should be emphasised, iron acquisition by bacteria and iron-deficient incubation test should be treated as a research tool but not aim.
2. In section 2.2 "Detection and identification of virulence genes" -
profiles of PCR are not clear, simplex or multiplex PCR was used? 94o C for 45 sec for annealing? For all reactions?
3. Results section: the origin of isolates from different farms is not important because the epidemic studies were not mentioned, this topic creates unnecessary confusion at work
4. A lot of results, hence the conclusion section at the end of the article is required.
Author Response
Question: The work is careful and well-documented, but the methodology used to testing of virulence is slightly outdated. The incubation test using an iron-deficient medium to estimate the carrying of iron-acquisition genes in bacteria and bacterial counts was known before the q/real-time PCR and CAS liquid assay era, these tests should be also used as a comparative method.
Answer: We agree Reviewer2’s comment: the laboratory tests (such as phenotype evaluation bacterial counts, and iron-deficient medium) that we have used in this study were slight-outdated methodology. One of main aims of this report was to develop our methods applicable routinely as the simple, field tests in bovine practice. To be able to utilize effectively the outdated devices commonly used in bovine clinics, our techniques would be required, despite of being slightly outdated.
Minor remarks:
Question: 1. In the Introduction section, the bovine mastitis problem should be emphasised, iron acquisition by bacteria and iron-deficient incubation test should be treated as a research tool but not aim.
Answer: In the revised version, the descriptions about bovine mastitis and iron-deficient incubation are added.
Question: 2. In section 2.2 "Detection and identification of virulence genes" - profiles of PCR are not clear, simplex or multiplex PCR was used? 94o C for 45 sec for annealing? For all reactions?
Answer: Sorry, the description of the annealing temperature was completely mistaken. Additionally, simplex PCR sequence was used for each virulence gene. In the revised version, the descriptions of our PCR method are corrected.
Question; 3. Results section: the origin of isolates from different farms is not important because the epidemic studies were not mentioned, this topic creates unnecessary confusion at work.
Answer: According to the comment in which the origin of isolates from different farms is not important, the sentences in lines 187-193 of previous manuscript “For farm A, from which 8 K. pneumoniae isolates were analyzed by PCR, KP0 was ………in which multiple isolates were analyzed by PCR (two to four isolates per farm) (Table S3).” are deleted. Additionally, according to this change, the descriptions of Discussion are slightly corrected or deleted.
Question; 4. A lot of results, hence the conclusion section at the end of the article is required.
Answer: According to this comment, the descriptions of “Conclusion” are added in the revised version.
Reviewer 3 Report
In the manuscript ID: microorganisms-1691426 the authors analyse the correlation between the PCR detection of iron uptake systems and the growth patterns exhibited by Klebsiella pneumoniae strains, isolated from bovine mild mastitis cases, in incubation tests in iron sufficient/deficient medium. Significant correspondences were mostly observed between the detected number of iron uptake genes and the growth observed after 3 days incubation in iron deficient medium, thus proposing the adopted experimental settings as promising and preliminary assay to investigate the state of iron uptake systems carriage by K. pneumoniae.
The paper is interesting and scientifically sounding. The introduction provides the necessary background to understand the topic and the importance of the present study; the methods are reported in detail, the results are clear and strongly corroborated by statistical analysis; the discussion explains all the obtained results and compares them to the previous findings present in literature.
There are just two minor corrections to check to make the manuscript even clearer:
-Is the annealing temperature (94°C) reported in line 113 correct? Please check.
-The study analysed 37 K. pneumoniae isolates; please check the isolates counts reported in lines 188, 189 comparing the numbers of strains isolated from Farm A and the remaining ones.
After solving these minor revisions, the paper can be published in “Microorganisms”.
MINOR COMMENTS
Line 29, please correct “an insufficient level”;
Line 32, please correct “transferrin”;
Line 47, please correct “The iucABCD gene cluster, which encodes”;
Line 112, please correct “consisting of the denaturation”;
Line 295, please delete “between farms”;
Line 428, please correct “can be developed by addition”;
Line 429, please correct “identifying the bacterial adhesion function”.
Author Response
The paper is interesting and scientifically sounding. The introduction provides the necessary background to understand the topic and the importance of the present study; the methods are reported in detail, the results are clear and strongly corroborated by statistical analysis; the discussion explains all the obtained results and compares them to the previous findings present in literature.
Answer: Thank you for your highly appreciation for this study.
Question: There are just two minor corrections to check to make the manuscript even clearer:
Answer: According to Reviewer3’s advice, the revised manuscript is made as follows.
Question: -Is the annealing temperature (94°C) reported in line 113 correct? Please check.
Answer: Sorry, the description of the annealing temperature was completely mistaken. In the revised version, the descriptions of our PCR method are corrected.
Question: -The study analysed 37 K. pneumoniae isolates; please check the isolates counts reported in lines 188, 189 comparing the numbers of strains isolated from Farm A and the remaining ones.
Answer: Sorry, the counts of samples were completely mistaken (“2 out of 29 (6.9%)” is right). However, because Reviewer2 requested the deletion of the sentences (in lines 187-193 of previous manuscript) including this sentence, this sentence is deleted in the revised version.
Question; MINOR COMMENTS
Answer: According to the following advice from Reviewer3, our manuscript is corrected.
Line 29, please correct “an insufficient level”;
Line 32, please correct “transferrin”;
Line 47, please correct “The iucABCD gene cluster, which encodes”;
Line 112, please correct “consisting of the denaturation”;
Line 295, please delete “between farms”;
Line 428, please correct “can be developed by addition”;
Line 429, please correct “identifying the bacterial adhesion function”.
Round 2
Reviewer 2 Report
Lines: 44-47 should be changed to a shorter form:
Enterobactin assists in intramammary infection [6,16,28], aerobactin contributes to the increased mastitis severity in cows [2,9,28], and yersiniabactin seems to prolong the intramammary survival of coliform bacteria, resulting in the chronicity of bovine mastitis [29].
Conclusions should be shorter (only the main conclusions from research without results).
Author Response
Thank you for your kindly advise.
Yellow-boxes highlight the corrected parts in revised version.
Question; Lines: 44-47 should be changed to a shorter form: Enterobactin assists in intramammary infection [6,16,28], aerobactin contributes to the increased mastitis severity in cows [2,9,28], and yersiniabactin seems to prolong the intramammary survival of coliform bacteria, resulting in the chronicity of bovine mastitis [29].
Answer: According to the advice, the previous sentences in lines 44-47 are replaced by this sentence in revised version.
Question; Conclusions should be shorter (only the main conclusions from research without results).
Answer: In section “Conclusion”, the words “(67.6%, both)” is deleted in lines 402-404 of the revised version. The sentence “The average number of PCR-positive iron-acquisition genes was 2.6, with a maximum of 5 in 13.5% K. pneumoniae isolates.” is also deleted. Additionally, the sentence “as the bacterial counts significantly increased between days 0 and 1 and were similar between days 2 and 3.” is deleted in lines 404-406 of the revised version.